# *Escherichia coli* Isolated from Cases of Colibacillosis in Russian Poultry Farms (Perm Krai): Sensitivity to Antibiotics and Bacteriocins

**DOI:** 10.3390/microorganisms8050741

**Published:** 2020-05-15

**Authors:** Marina V. Kuznetsova, Julia S. Gizatullina, Larisa Yu. Nesterova, Marjanca Starčič Erjavec

**Affiliations:** 1Institute of Ecology and Genetics of Microorganisms, Ural Branch of the Russian Academy of Sciences, Goleva street 13, 614081 Perm, Russia; mar@iegm.ru (M.V.K.); gizatullina.julia@yandex.ru (J.S.G.); larisa.nesterova@bk.ru (L.Y.N.); 2Department of Biology, Biotechnical Faculty, University of Ljubljana, Jamnikarjeva 101, 1000 Ljubljana, Slovenia

**Keywords:** avian *Escherichia coli*, antibiotics, bacteriocins, colicins, microcins

## Abstract

*Escherichia coli* strains isolated from case of colibacillosis in Russian poultry farms in the region of Perm Krai were analyzed for their sensitivity to main antibiotics and bacteriocins. Sensitivity profiles for 9 antibiotics and 20 bacteriocins were determined with the disc diffusion method and the overlay test, respectively. Further, with the PCR the presence of several *bla* and integron 1 genes was revealed and the phylogenetic group for each strain determined. Among the 28 studied *E. coli* strains 85.7% were resistant to at least three antibiotics, 53.6% to five or more drugs, and 10.7% to eight antibiotics. PCR revealed that the *bla*_TEM_ gene was harbored by 71.4% of strains and the *bla*_CTX-M_ gene by 53.6% of strains. The class 1 integrons were found in 28.6% of strains. All of the studied strains were insensitive to ten or more bacteriocins. More than 90% of the studied strains were insensitive to pore-forming colicins of group A and B colicins, while 60.7% were insensitive to colicins with DNase and RNase activity. All of the analyzed strains were insensitive to at least two of the tested microcins. Neither the antibiotic resistance profile nor the bacteriocin resistance profile correlated with phylogenetic group of the strains. Thus, the studied strains were shown to possess high levels of multiple resistance to antibiotics and insensitivity to bacteriocins.

## 1. Introduction

*Escherichia coli* are part of the mammalian and human intestinal microbiota. However, they are also widely distributed in the environment. Among the *E. coli* strains, not just commensal, but also pathogenic, intestinal pathogenic *E. coli* strains (IPEC) and extraintestinal pathogenic *E. coli* (ExPEC), can be found. ExPEC can be divided into uropathogenic (UPEC), sepsis-associated (SEPEC) and meningitis-associated (in newborns) strains (NMEC) [1]. In birds, the *E. coli* also has “two faces”; it is found in the intestinal microbiota of healthy poultry, but also associated with extraintestinal diseases [2,3]. The avian pathogenic *E. coli* (APEC) is the causative agent of colibacillosis and septicemia in birds that leads to localized inflammation, most commonly presenting as perihepatitis, airsacculitis, and/or pericarditis and lymphocytic depletion of the bursa and thymus. Colisepticemia is a serious life-threatening condition, which is associated with acute onset of very generalized clinical signs of sickness, such as listlessness, depression, weakness, loss of appetite, and sudden death of birds. APEC is the main cause of mortality in poultry farms [4]. According to the Canadian Antimicrobial Resistance Surveillance System report [5], the morbidity rate with *Escherichia* in Canadian poultry was approaching 96% in 2016. The veterinary reports of the Russian Federation for 5 years (2009–2014) provided by the FGBI Veterinary Centre revealed significant problems of the national commercial poultry farming. Based on the data available it is evident that bacterial diseases (salmonellosis, colibacillosis, ornithosis, and pasteurellosis) account for most outbreaks. In most cases the diseases are caused by violations of poultry management and feeding standards [6]. In Russia, colibacillosis accounts for between 60% and 88% of all poultry infections [7]. Colibacillosis hence leads to considerable economic losses in poultry farming [4,8]. In the Russian Federation 83% of poultry farming is accounted to conventional agricultural farms/organizations and produces almost 5 million tons of poultry in live weight per year [9].

The presence of similar phylogroups, serotypes and genetic determinants among representatives of IPEC, ExPEC, and APEC suggests that the latter group may be a reservoir of virulence genes and may also pose a zoonotic risk [10,11], as there is a possibility that the animal bacteria can be transmitted to humans either through direct contact or through the use of poultry products [12].

Over the last 50 years, the use of antibiotics in combination with biological safety measures has contributed to a significant increase in poultry farming [13]. In most poultry farms in Russia and abroad, antimicrobial agents are added to animal feed on an ongoing basis. In Russia, as well as in many other countries, the main drugs used to prevent colibacillosis in the parent flocks and broilers in conventional agricultural farms are cephalosporins and fluoroquinolones [14]. Although the drugs used are veterinary, they belong to the same antibiotic groups as antimicrobial drugs intended for the treatment of serious human infections, including those classified as antimicrobial agents of III and II (“important” and “very important”) categories and even category I (“critically important”), e.g., cephalosporins [15].

Currently, an increase in the number of antibiotic-resistant bacterial infections is observed. Additionally, this increase has already reached such a high level that it has become one of the main public health problems. Multicomponent antimicrobials have been continuously used as growth promoters and for the prevention and treatment of infectious diseases in poultry, leading to the emergence of ecovars of resistant bacteria. Numerous studies show that, among the *E. coli* isolates circulating in poultry farms, resistance to tetracycline, nalidixic acid, ampicillin, amoxicillin, streptomycin, trimethoprim, and cotrimoxazole appears to be the most common and more than half of the aetiopathogens are multiresistant [16,17,18,19,20]. The release of antibiotic-resistant strains into the environment by animal waste products extends the “resistance reservoir” present in the microbiome of natural biotopes [21].

In this context, the World Health Organization (WHO) suggested a new strategy for combating infectious diseases in poultry, namely the use of probiotic drugs. Probiotics based on microbial living cells, including *E. coli* cultures, are used to form a colonization barrier in the host gut, often based on bacteriocin production by the probiotic microorganism. Bacteriocins are antimicrobial peptides that are active against taxonomically closely related microorganisms. *E. coli* produces two types of bacteriocins—colicins and microcins. However, these probiotic drugs may not be effective enough because of the intrinsic bacterial insensitivity to bacteriocins associated with the bacteriocin’s receptor modifications or translocation systems [22,23]. Considering that both antibiotic and probiotic preparations are actively used in agriculture, including on poultry farms, monitoring of microbial susceptibility to antibiotics and bacteriocins among *E. coli* strains circulating in poultry is important. Therefore, the aim of the work was to investigate the sensitivity to antibacterial drugs of main groups that are used to prevent colibacillosis on Russian poultry farms and 20 most prominent bacteriocins among *E. coli* strains obtained from poultry farms with cases of colibacillosis in the region of Perm Krai. To our knowledge, this is the first study including bacteriocin sensitivity testing for Russian *E. coli* strains.

## 2. Materials and Methods

### 2.1. Farms and Samples

Three different conventional broiler and parent flock large farms located in the Perm region (Russia) in the period April 2016 to March 2018 were included in the study (Table 1).

Samples (organs with typical lesions of colisepticemia) were collected from randomly chosen 50 sick birds (aged 7–36 and 177–364 days) showing clinical signs of colibacillosis (watery diarrhea, weakness, anorexia, weight loss, etc.) by farm veterinarians. Liver, heart, spleen, and lungs from each bird; in some cases, kidneys (22 samples), oviduct (18 samples), trachea (5 samples), and bones (5 samples), all together 250 samples were taken by farm veterinarians. All organ samples were collected separately in sterile tubes or boxes and sent for examination. Unfortunately, we were not provided with the number of all sick birds, neither with an estimation of the proportion of sick birds on the farms by the farm veterinarians.

### 2.2. Bacterial Strains

In order to isolate the bacteria an incision was made in the parenchymal organ of the bird and the inner part of the organ was sown directly onto the surface of the MacConkey agar (Sigma-Aldrich, St. Louis, MO, USA) and sheep blood agar medium. Sheep blood agar was used in parallel with MacConkey agar as an enrichment medium in the case of a low bacterial concentration in the organ. The bone was broken, the bone marrow was taken with an inoculation loop and the seeding was done also on both types of agar medium. All further analyzed typical *E. coli* colonies (pink colored, transparent smooth and raised colonies or dark red colonies with pronounced metallic luster), one colony per each sampled organ with positive seeding, thus 196 colonies, were taken from the MacConkey agar and subcultured onto MacConkey agar again. Gram staining (Gram-negative rods) and biochemical tests: Kligler iron agar (decomposes glucose and lactose to acid and gas, without hydrogen sulfide-yellow color with interruptions); Simmons citrate agar (does not use citrate-no change in the color); and the catalase (positive) and oxidase test (negative) were used for preliminary identification of bacteria (Table 1—column *N* of samples positive for *E. coli*). Finally, for confirmation of *E. coli* by the ENTEROtest 16 diagnostic system (Erba Lachema s.r.o., Czech Republic) with the russified version of the Micro-La-test “Microb-2” (microbiological system monitoring “Microbe-2”, SMMM-2) computer program, one randomly chosen isolate from each bird, thus 50 isolates, were taken. Isolates with an “acceptable” rating and a match rate of more than 94% were confirmed as *E. coli*. All 50 by ENTEROtest 16 analyzed isolates were confirmed to be *E. coli*, so all were taken for the final differentiation between clonal and non-clonal *E. coli* with ERIC-PCR [24,25]. So, finally, 28 non-clonal *E. coli* strains were collected and further analyzed in this study (Table 2).

### 2.3. Antibiotic Sensitivity Testing

Sensitivity to antibiotics ampicillin, cefotaxime, ceftazidime, amikacin, gentamicin, tetracycline, ciprofloxacin, levofloxacin, and co-trimoxazole (trimethoprim/sulfamethoxazole) was determined by the disk diffusion method (EUCAST 2018, MACMAX, version 2018-03). 

### 2.4. Bacteriocin Sensitivity Testing

Studied *E. coli* strains were screened for bacteriocin sensitivity using a collection of indicator strains producing colicins and microcins (BZB collection, University of Ljubljana, Slovenia). The effects of group A and B colicins, as well as class I and II microcins were evaluated [23]. Colonies of the bacteriocinogenic indicator strains were grown on Luria-Bertani agar (LB-agar; Amresco, Solon, OH, USA) in a Petri dish (9 cm) for 24 h. The grown colonies were treated with chloroform vapors for 15 min. A suspension of the analyzed *E. coli* strain (standardized to 2.0 according to MacFarland) was added to the melted (46 °C) 0.6% LB soft agar, mixed, and poured over the prepared vaporized colonies. The plates were then incubated for 24 h at 37 °C. The next day, the plates were examined and growth inhibition zones were measured. Inhibition zones of 1 mm or more were considered to reveal sensitive *E. coli* strains. Using the same method, except that the strains were grown as colonies and the DH5α strain, known to be sensitive to bacteriocins, was used as the overlaid indicator strain, all studied strains were also tested for their ability to produce bacteriocins themselves. 

### 2.5. Detection of Antibiotic Resistance and Integron Genes

The selection of antibiotic resistance genes tested was based on literature [26,27,28,29]. The strains were tested for the *bla*_TEM_, *bla*_SHV_, *bla*_OXA_, *bla*_CTX-M_, and *bla*_CMY_ β-lactamase-encoding genes by PCR using universal primers for the TEM, SHV, OXA, CTX-M, and CMY families. Multiplex PCR for the molecular detection of resistant genes specific for ampicillin and amoxicillin-clavulanic acid (*bla**_TEM_*, *bla**_SHV_*, and *bla**_OXA_*) was carried out using three primer pairs [26]. The CTX-M cluster was detected according to [27] and CMY-2 cluster as the most frequently reported AmpC β-lactamase was detected according to [28]. The occurrence of class 1 integrons was tested using 5′CS/3′CS primers complementary to conserved integrons region between the 5′ conserved segment and 3′ conserved segment as previously described [29]. Primers were synthesized in Evrogen LLC (Moscow, Russia; Appendix A). Amplification protocols were carried out according to the authors’ recommendations [26,27,28,29]. Amplification was performed on a DNA Engine Dyad Thermal Cycler (Bio-Rad, Foster City, CA, USA). Visualization of bands and data documentation were carried out using the Gel-Doc XR gel documentation system (Bio-Rad, Foster City, CA, USA).

### 2.6. Phylotyping

Phylotyping was performed by a multiplex PCR according to Clermont et al. (2013) [30].

### 2.7. Statistical Analysis

Statistical analysis was performed using Microsoft Office XP Excel 2013. To study the relationship between two parameters, a nonparametric correlation coefficient—Spearman coefficient (rs) was calculated; statistical significance level of *p* = 0.05 was used.

## 3. Results and Discussion

### 3.1. Antibiotic Sensitivity and Presence of Antibiotic Resistance and Integron Genes

The strains analyzed showed a high antimicrobial resistance (Table 2). 

The highest number of resistant strains was found for the resistance to ampicillin and co-trimoxazole (trimethoprim/sulfamethoxazole), as more than 80% of strains exhibited this resistance (Table 3). High levels of multidrug resistance were observed, as 85.7% of strains were resistant to at least three antibiotics, 64.3% of cultures showed resistance to more than three antibiotics, 53.6%—to five or more drugs, and 10.7%—to even eight antibiotics. The performed study included the main drugs’ groups used to prevent colibacilosis in the parent flocks (cephalosporins) and in broilers (fluoroquinolones) in Russian conventional agricultural farms. In addition, these antibiotics are clinically important as they are also commonly used in the Russian medical system for treatment of human diseases. A study conducted in Canada by Boulianne M. et al. (2016) showed a significant correlation between the usage of Ceftiofur, an antibiotic used in veterinary medicine, and the resistance of *E. coli* isolates to other beta-lactam antibiotics [17]. Additionally, in our study most *E. coli* strains turned out to be insensitive to for human medicine clinically important antibiotics.

In this study, we analyzed genes related to resistance to β-lactam antibiotics belonging to five different types of β-lactamases TEM, SHV, OXA, CTX-M, and including CMY-2, which belongs to a small family of plasmid-mediated AmpC-like enzymes. Specific amplification of the *bla*_TEM_ gene was detected in 71.4% of strains, *bla*_CTX-M_ in 53.6% and *bla*_SHV_ was detected in one studied strain (3.6%). Genes of serine beta-lactamases of classes C (*bla*_CMY_) and D (*bla*_OXA_) were not found in any of the strains analyzed. The mechanisms of resistance against beta-lactam antibiotics are known to be very diverse. This is also clearly visible in our performed genetic background analysis. Cephalosporin resistance is attributed to other mechanisms, for example, PBP-modification or change of permeability [31,32]. The study of Choi and Lee (2019) showed porin-mediated antibiotic resistance: the *ompF* and *ompC* mutants showed significantly increased resistance to several β-lactam antibiotics, including cephalosporins [33]. In addition, *E. coli* resistance can be due to the presence of other specific, less prevalent ESBL genes, for example *bla*_VEB_, which was not investigated in our research [34,35]. Further, it would be interesting to obtain the complete genome nucleotide sequence of the strain in our studied collection that was phenotypically resistant to ceftazidime, but was not found to be positive in the performed PCR reactions (strain number 4). In eight studied strains integron fragments were detected, which varied in size from 800 to 2000 bp ( Appendix A).

The most common mechanism for the resistance to beta-lactam antibiotics in *E. coli* is the production of broad and extended spectrum beta-lactamases (ESBL), which confer the resistance to several antibiotics, including fourth-generation cephalosporins and monobactams. Beta-lactamases are a broad class of enzymes whose genes are often localized on plasmids, which contributes to their horizontal transfer between bacteria and determines the epidemiological importance. Koga et al. (2015) discovered a large number of ESBL-producing isolates (mainly CTX-M enzyme) after examining more than 200 *E. coli* strains isolated from chicken carcasses [36]. The *bla*_TEM_ and *bla*_CTX-M_ genes circulate most frequently among the APEC strains found on poultry farms in Egypt and Brazil [37,38]. The same was observed also in our study, where these two genes were found to be present in 71.4% and 53.6%, respectively. In recent years, there has been an increase in the prevalence of *E. coli* producing CTX-M type ESBL, with CTX-M-1 producer strains being the most common among animals, while CTX-M-14 and CTX-M-15 producers are more common among humans [39]. Our study showed that the predominant enzymes in *E. coli* strains circulating in large poultry farms were TEM and CTX-M type beta-lactamases. The combination of *bla*_TEM_+ *bla*_CTX-M_ genes was detected in 13 studied strains (46.4%). Further, our study also showed that strains possessing class 1 integrons, possessed also genes for the resistance to ampicillin and/or cephalosporins. Thus, our data once again showed that avian *E. coli* strains are a reservoir of resistance genes that can be transmitted due to mobile genetic elements.

### 3.2. Bacteriocin Sensitivity and Production 

All of the studied strains were found to be insensitive to at least ten bacteriocins, 11 strains (39.3%)—to more than fifteen, and 2 strains were insensitive to all bacteriocins analyzed (Table 2 and Table 4). 

Pore-forming bacteriocins (colicins A, K, N, S4, E1, B, Ia, Ib, D, and M), except colicin K, manifested least efficacy against studied *E. coli* strains, as more than 90% of strains were insensitive to them (Table 4). Colicins with DNase and RNase activity (colicins E2, E3, E4, E5, E6, E7, and E8J) were more effective, while the insensitivity to the other nuclease colicins was revealed in 60% of strains. Levels of insensitivity to microcins of both classes were also high (at least 78.6%). 

Bacteriocin production was revealed for 71.4% of the studied strains. Statistical analysis showed that the bacteriocin insensitivity was not associated with the bacteriocin production.

### 3.3. Associations of Antibiotic (in) Sensitivity and Bacteriocin (in) Sensitivity and Phylogenetic Groups

The prevalence of phylogroups among the studied strains was as follows: B1—8 (28.6%) strains, B2 and E—4 (14.3%) strains, A, C, and F—2 (7.1%) strains, D—1 (3.6%) strain, and U (unidentified)—5 (17.9%) strains. There were no associations between the antibiotic resistance and the strains’ phylogroups, or associations between bacteriocin insensitivity or production with the phylogroups.

A weak negative association of antibiotic insensitivity and bacteriocin insensitivity was revealed by calculation of the Spearman coefficient (rs), i.e., strains being sensitive to antibiotics were more often insensitive to a large range of bacteriocins (rs = −0.27). However, the effectiveness of colicins with DNase activity did not depend on the strain insensitivity to antibiotics, notably this type of colicins was effective even for multi-resistant bacteria.

Among bacteriocinogenic strains, 60% were resistant to five or more antibiotics, and among non-bacteriocinogenic strains there were only 37.5%. However, bacteriocin resistance in both groups was approximately similar (55% and 50% of strains being resistant to 15 or more bacteriocins, respectively).

According to da Rocha et al. (2002), 87.3% of the APEC strains isolated from colibacillosis outbreaks in industrial plants in Brazil were bacteriocinogenic [12]. In our study, 71.4% of the strains were found to produce bacteriocins. The biological characterization of bacteriocinogenic strains is important, because bacteriocins are often plasmid-encoded and such plasmids often carry other virulence and resistance genes. A good example of such a plasmid is the ColBM plasmid (pAPEC-O103-ColBM0), which contains genes for multidrug resistance, the microcin V gene, and pathogenicity islands associated with the ability of APEC to cause septicemia in chicken and meningitis-associated bacteremia in a rat model [38].

Bacteriocins were proposed as new potential antimicrobial agents, but the prevalence of bacteriocin insensitivity hinders their potential. With regard to bacteriocin insensitivity, a study by Budič M. et al. (2011) showed that colicin E7 was most effective against human clinical *E. coli*, namely only 13% of the strains analyzed were colicin E7 insensitive, while 32%, 33%, 43% and 53% were insensitive to colicin E6, K, M, and E1, respectively [23]. In our study the most effective was colicin E4—25% were colicin E4 insensitive, where 53.6% of tested strains were colicin E7 insensitive. These kinds of data shows that the most effective bacteriocins can differ for different subsets of *E. coli*. Therefore, probiotics based on bacteriocin production should be carefully chosen and administered only if proven to be active on a certain *E. coli* subset. 

## 4. Conclusions

This work has shown that the *E. coli* strains circulating in studied Russian poultry farms with cases of colibacillosis are resistant to a large number of antibiotics tested and that more than half of the studied strains are multidrug-resistant. The strains analyzed produced TEM and CTX-M type beta-lactamases, most of them were associated with class 1 integrons. In addition, it was found that the studied strains were highly insensitive also to bacteriocins. Apparently, the antimicrobial potential of probiotic preparations based on bacteriocins intended for poultry farming can only be realized in a combination of several types of bacteriocins with different mechanisms of action.

## 5. Final Remarks

As the studied *E. coli* strain were isolated from organs of sick birds from colibacillosis cases as determined by field veterinarians on the farms, it can be assumed that the studied *E. coli* strains were actually APEC, but the golden standard experiment (inoculation into embryos or one-day-old chicken) to confirm their pathogenicity was not performed. Further, the golden standard method for discrimination of non-clonal isolates, the PFGE or AFLP, was not performed. Instead of the simple and cost-effective genotyping method of ERIC-PCR was used, which was demonstrated to have in determining the genetic diversity the same discriminative power as AFLP [25]. This study, even though only 28 *E. coli* strains were included, revealed that probiotic preparations based on a single bacteriocin are probably not going to be effective enough in replacing classical antibiotics, as the insensitivity to bacteriocins was even higher among the studied strains as the insensitivity to most used antibiotics. Further, the study revealed the importance of future monitoring of a larger number of *E. coli* strains for insensitivity to antibiotics and bacteriocins to be able to limit/prevent poultry colibacillosis in farms.

## Figures and Tables

**Table 1 microorganisms-08-00741-t001:** Agricultural farms and samples.

Year	Agricultural Farm					
	Farm 1 (Broiler)		Farm 2 (Parent Flock)		Farm 3 (Broiler)	
	*N* of Samples (from *N* of Birds)	*N* of Samples Positive for *E. coli*	*N* of Samples (from *N* of Birds)	*N* of Samples Positive for *E. coli*	*N* of Samples (from *N* of Birds)	*N* of Samples Positive for *E. coli*
2016	43 (6)	36	21 (3)	18	25 (3)	15
2017	26 (4)	22	29 (12)	26	27 (5)	24
2018	18 (2)	13	35 (11)	22	26 (4)	20

**Table 2 microorganisms-08-00741-t002:** All obtained data of the studied strains.

Strain	Isolation Month, Year	Poultry Farm	Organ Sample	Phylogroup	Bacteriocin Production (+)	Insensitive to Antibiotics (+)									Presence of ESBL and Class 1 Integron Genes (+)						Insensitive to Bacteriocins (+)																			
						Amikacin	Gentamicin	Tetracycline	Ciprofloxacin	Levofloxacin	Co-trimoxazole	Ampicillin	Cefotaxime	Ceftazidime	TEM	SHV	OXA	CTX-M	CMY	3/5cs	ColA	ColB	ColD	ColE1	ColE2	ColE3	ColE4	ColE5	ColE6	ColE7	ColIa	ColIb	ColK	ColN	mccB17	ColM	ColS4	mccC7	ColV	ColE8J
4	Apr 16	1	Liver	U	-	-	-	+	-	-	-	-	-	+	-	-	-	-	-	-	+	+	+	+	+	+	+	+	+	+	+	+	+	+	+	+	+	+	+	+
5	Apr 16	2	Kidney	B1	+	-	-	+	-	-	-	-	-	-	-	-	-	-	-	-	+	+	+	+	+	+	+	+	+	+	+	+	-	+	-	+	+	+	+	+
7	May 16	1	Liver	B1	-	-	-	+	-	-	+	+	-	-	+	-	-	+	-	+	+	+	+	+	-	+	-	+	-	+	+	+	+	+	+	+	+	+	+	-
9	May 16	1	Lungs	E	+	-	+	+	-	-	+	+	+	+	+	-	-	+	-	-	+	+	+	+	-	-	-	+	-	+	+	+	+	+	-	+	+	+	+	-
11	Sept 17	2	Liver	B2	-	-	+	-	-	-	+	+	-	-	+	-	-	+	-	-	+	+	+	+	+	+	-	-	-	+	+	+	+	+	+	+	+	+	+	-
12	Sept 17	2	Oviduct	C	+	-	+	+	+	+	+	+	-	-	+	-	-	+	-	-	+	+	+	+	+	-	-	+	-	-	+	+	-	+	+	+	+	+	+	+
15	Oct 17	2	Spleen	B2	+	-	-	-	-	-	-	-	-	-	-	-	-	+	-	-	+	+	+	+	-	+	-	+	+	-	+	+	+	+		+	+	+	+	-
16	Oct 17	2	Lungs	D	-	-	-	+	-	-	+	+	-	-	+	-	-	-	-	-	+	+	+		-	-	-	+	-	-	+	+	-	+	+	+	+	+	+	-
17	Oct 17	2	Oviduct	F	+	-	-	+	-	-	+	+	-	-	+	-	-	-	-	-	+	+	+	+	+	+	-	+	+	-	+	+	+	+	+	+	+	+	+	+
20	Oct 17	2	Liver	E	+	-	-	-	+	+	+	+	-	-	+	-	-	+	-	+	+	+	+	+	+	+	+	+	+	+	+	+	+	+	+	+	+	+	+	+
21	Oct 17	1	Lungs	F	-	-	-	+	-	-	+	+	-	-	+	-	-	+	-	+	+	+	+	+	-	+	-	+	-	+	+	+	+	+	+	+	+	+	+	-
24	Nov 17	3	Heart	B1	+	-	+	+	+	+	+	+	+	+	+	-	-	-	-	-	+	+	+	+	+	-	-	-	-	+	+	+	-	+	+	+	+	+	+	-
26	Nov 17	3	Lungs	E	+	-	+	+	+	+	+	+	+	+	+	-	-	+	-	-	+	+	+	+	+	+	-	-	-	-	+	+	+	+	+	+	+	+	+	-
29	Nov 17	3	Liver	E	+	+	+	+	+	+	+	+	-	-	+	-	-	-	-	-	+	+	+	+	+	-	-	+	-	+	+	+	+	+	+	+	-	+	-	+
31	Nov 17	3	Liver	B1	-	-	-	+	+	+	+	+	-	-	+	-	-	+	-	+	+	+	+	+	-	-	-	-	-	-	+	+	-	+	+	+	+	+	+	-
33	Dec 17	2	Heart	B1	+	+	+	+	+	+	+	+	-	-	+	-	-	+	-	+	+	+	+	+	-	+	-	+	+	-	+	+	+	+	+	+	+	+	+	-
35	Dec 17	1	Spleen	B2	+	-	+	+	+	+	+	+	-	-	+	-	-	-	-	-	+	+	+	-	-	-	-	-	-	-	+	+	-	+	+	+	+	+	+	-
36	Dec 17	1	Liver	B1	+	-	-	+	+	+	-	+	-	-	+	-	-	+	-	-	-	+	+	+	+	+	+	+	+	+	+	+	-	+	+	+	+	+	+	-
37	Feb 18	1	Liver	U	+	+	+	+	+	-	+	+	+	+	-	-	-	+	-	-	+	+	+	-	-	-	+	+	-	-	+	+	+	+	-	+	-	+	+	-
41	Feb 18	1	Lungs	U	-	-	+	+	+	+	+	-	-	-	-	-	-	-	-	-	+	+	+	+	-	+	-	-	-	+	+	+	-	+	+	+	+	+	+	-
42	Mar 18	2	Kidney	B1	+	-	-	+	-	-	+	+	-	-	-	-	-	-	-	-	+	+	+	+	-	-	-	+	-	-	+	+	-	+	+	+	+	+	+	-
43	Mar 18	2	Lungs	B2	+	-	-	-	-	-	-	-	-	-	-	-	-	-	-	-	+	+	+	+	+	-	-	-	-	-	+	+	+	+	-	+	+	+	+	+
45	Mar 18	3	Trachea	C	-	-	-	+	+	+	+	+	-	-	-	+	-	-	-	+	+	+	+	+	-	-	-	-	-	+	+	+	-	+	+	+	+	+	+	-
46	Mar 18	3	Liver	B1	+	-	-	+	+	+	+	+	-	-	+	-	-	-	-	-	+	+	+	+	-	-	+	-	+	-	+	+	-	+	+	+	+	+	+	-
11/16	May 16	2	Liver	A	+	-	+	+	-	-	+	+	+	+	+	-	-	+	-	-	+	+	+	+	+	+	+	+	+	+	+	+	+	+	-	+	+	+	+	+
13/16	May 16	2	Kidney	A	+	-	+	+	-	-	+	+	+	+	+	-	-	+	-	+	+	+	+		-	+	-	+	-	+	+	-	+	+	+	+	+	+	+	-
14/16	May 16	2	Liver	U	+	-	+	-	-	-	+	+	+	+	+	-	-	+	-	-	+	+	+	+	-	+	-	-	-	+	+	+	-	+	+	+	+	+	+	-
20/16	May 16	2	Lungs	U	+	-	-	-	+	+	+	+	-	-	+	-	-	-	-	+	+	+	+	+	+	-	-	-	-	-	+	+	-	+	+	+	+	+	+	-

**Table 3 microorganisms-08-00741-t003:** Prevalence of resistance to antibiotics among studied strains.

Antibiotic	Antibiotic Group	Inhibition Zone for Resistant Isolates (mm) (Disk Diffusion Method)	Disk Antibiotic Content (µg)	Resistant Strains (%)
Amikacin	Aminoglycosides	<15	30	10.7
Gentamicin	Aminoglycosides	<14	10	46.4
Tetracycline	Tetracyclines	<12	30	78.6
Ciprofloxacin	Quinolones	<24	5	50.0
Levofloxacin	Quinolones	<19	5	46.4
Co-trimoxazole	Trimethoprim/Sulphonamides	<11	1.25/23.75	82.1
Ampicillin	β-Lactams	<14	10	82.1
Cefotaxime	β-Lactams	<17	5	42.9
Ceftazidime	β-Lactams	<19	10	39.3

**Table 4 microorganisms-08-00741-t004:** Prevalence of insensitivity to bacteriocins among studied strains.

Bacteriocin	Bacteriocin Group	Insensitive Strains (%)
A	Group A colicins	96.4
K	Group A colicins	53.6
N	Group A colicins	100
S4	Group A colicins	92.9
E1	Group A colicins	85.7
E2	Group A colicins	46.4
E3	Group A colicins	53.6
E4	Group A colicins	25.0
E5	Group A colicins	60.7
E6	Group A colicins	32.1
E7	Group A colicins	53.6
E8J	Group A colicins	28.6
B	Group B colicins	100
Ia	Group B colicins	100
Ib	Group B colicins	96.4
D	Group B colicins	100
M	Group B colicins	100
B17	Class I microcins	78.6
C7	Class I microcins	100
V	Class II microcins	96.4

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
