# Peer review of "Escherichia coli Isolated from Cases of Colibacillosis in Russian Poultry Farms (Perm Krai): Sensitivity to Antibiotics and Bacteriocins"

_microorganisms, 2020, doi:10.3390/microorganisms8050741_

Round 1
Reviewer 1 Report
The manuscript includes a study on the prevalence of antibiotic- and bacteriocin-resistant strains of Escherichia coli that caused colibacillosis in poultry farms. The methodology used is correct, the results are clearly presented and updated references are included.
I recommend the publication of the article.
Just a few typographical errors:
lines 166 & 174, et al. in cursive
line 186, ... strains (s) are ...
Author Response
The authors are deeply thankful for all the reviewer’s efforts and comments and we do appreciate them very much. All reviewer's comments were incorporated into the revised version of the manuscript. All et al. (5-times used) are now in italics. The typographical error ... strains (s) are … is now corrected.
Reviewer 2 Report
General comments
The manuscript by Marina Kuznetsova et al. examines the antimicrobial resistance and the sensitivity to bacteriocins among 28 E. coli isolates obtained from diseased broilers.
The manuscript is well written, the subject has public health and animal health importance, and data regarding broilers raised in Russia was not previously published in scientific literature. In a globalized world, we will not succeed in preventing the emergence and dissemination of resistant strains if we were unable to draw a truthful epidemiological puzzle. Thus, information collected in a maximum diversity of biomes and geographies is precious.
However, this manuscript refers to a very limited investigation. Neither the methodologies used in this study nor its statistical power allow any broad conclusion on both molecular characterization (which is basic) and the epidemiological importance (due to a very limited number of strains). So, I cannot support publication of the paper as it is written. Authors must perform additional work in order to obtain more solid results and the manuscript (mainly the discussion section) should be deeply shortened.
I trust that the authors will find the following comments and suggestions helpful towards the improvement of their research/manuscript.
#1 The authors claim the study of 28 Avian Pathogenic Escherichia coli (APEC) isolates without revealing any information about their serotype virulence. Although, phylotyping have been performed, its inoculation into embryos or one-day-old chicks is still considered the golden standard to confirm their pathogenicity. Furthermore, several genes that currently considered as the minimal predictors of APEC virulence (e.g., hlyF, iroN, iss, iutA, and ompT) should be screened.
#2. Secondly, if the study was aimed to assess the sensitivity to antibacterial drugs and bacteriocins of APEC, how was the necessary sample size determined? Are the authors convinced that 28 isolates could be sufficiently representative of strains circulating in Russian poultry farms? More E. coli isolates should be included, and all the isolates must be tested against a broader panel of clinical and public health relevant antibiotics. With this information authors could draw stronger conclusions about spread of MDR E. coli in Russian poultry flocks and about the animal health and zoonotic risks they represent. This additional data will be also helpful to assist field veterinarians when they need to prescribe empirical antimicrobial treatments (better accuracy and prudence).
#3. Thirdly, resistance is not a static phenomenon; it evolves over time and follows the selective pressure. It would be very enlightening if the authors were able to introduce data on antimicrobial use in the sampled farms. Without this information a very limited discussion could be made about selective factors operating in the Russian poultry environment.
#4. Why were bla β-lactamase genes not sequenced in this investigation? Without this, no reliable comparison can be made with either the results obtained in other regions and countries nor with future investigations in Russia.
#5, Enterobacterial Repetitive Intergenic Consensus (ERIC) PCR is a simple and cost-effective genotyping technology for discriminating different types of strains; however, this method is far from being accepted as a reference or standard. A more established method, such PFGE or MLST, would be advisable to be performed. At least address this “weakness” at the discussion section.
Specific comments
Materials and Methods
The subsection “Bacterial strains” is incomplete. The following questions or omissions must be addressed:
How many poultry farms were involved? Is there any other information available pertaining to the production system (conventional, domestic, etc) and antimicrobial use in the study farms? What are the most frequent pathologies?
All the isolates were recovered from extraintestinal organs? What culture media was used for the initial purification? How many isolates were selected per sample?
The authors are reminded that Materials and Methods sections are supposed to provide enough information to allow a clear reading of the research. In addition, this information is crucial to discuss the results of this study regarding antimicrobial resistance frequencies.
Please remove Table 1 (primers for PCR-detection of ESBL-genes and class 1 integrons) to supplementary materials.
Results
Your results section is a repetition of data that are in the tables, and therefore must be drastically cut back. Results must be displayed in a good table complemented with one or two sentences that point out the big purpose of or conclusion from the table.
Please remove Figure 1 to supplementary materials and add information about the molecular weight in lanes 1-28 (samples) and which lane correspond to the positive control.
Discussion
Too long and largely a literature review or a mere comparison with the results obtained in few similar investigations. Instead, could the authors present possible explanations for the results obtained? Lines 207 to 213 are the only ones that focus their discussion on their observations.
Author Response
The authors are deeply thankful for all the reviewer’s efforts and comments and we do appreciate them very much. The comments were used to revise the manuscript accordingly and we do hope that the revised manuscript is now improved and worth to be published.
Reviewer’s Comment #1: confirmation of strains’ pathogenicity
All analyzed strains originated from colibacillosis cases as determined by field veterinarians on the farms. All analyzed E. coli were obtained from different organs of dead broilers as collected by veterinarians. The data on the origin of the strains are now given in the Supplementary Table 1, which shows all the gathered data on the analyzed strains. Since all strains analyzed are isolates from different organs of dead animals and the pathogenicity of the analyzed strains was detected in vivo on farms by veterinarians, we believe that no further in vitro experiments are necessary to confirm the pathogenicity of the strains.
Reviewer’s Comment #2: aim of the study and the number of strains
We would like to stress that the submitted manuscript is a short communication oriented towards analysis of strains obtained in colibacillosis cases on three Russian farms of Perm Krai aimed at insensitivity to antibiotics and bacteriocins of the isolated strains. We do agree that the number is rather small, so we changed the title of the manuscript and added (Perm Krai) into the title.
Reviewer’s Comment #3: data on antimicrobial use in the sampled farms and selective factors and evolution
Unfortunately we do not have disclosure rights to list the antibiotics used on the farms. However, there was a 14-years long monitoring study including clinical and veterinary antibiotics on one poultry farm and these data was recently published in the article: Kuznetsova, M.V.; Afanasievskaya, E.V.; Pokatilova, M.O.; Kruglova, A.A.; Gorovitz, E.S. Diversity and antibiotic resistance of enterobacteria isolated from broilers in a poultry farm of Perm krai: а 14-year study. Agricul. Biol. 2019, 54, 754-766. This reference is now also used in the revised manuscript.
And once again we would like to point to the aim of our manuscript. Revealing selective factors and evolution dynamics of antimicrobial resistance is out of the scope.
Reviewer’s Comment #4: reliable comparison with results obtained in other regions and countries
As written in the Material and method section several antibiotic resistance genes were screened in PCR-reactions with protocols and primers that are established as molecular tools for detection of antibiotic resistance genes. All the references are cited. Hence, the obtained data can be used in reliable comparison.
Reviewer’s Comment #5: ERIC-PCR and more established methods for strain discrimination
Beside PFGE and MLST, AFLP is also an established method for strain discrimination and in one of our previous published studies Fajs, L.; Jelen, M.; Borić, M.; Đapa, T.; Žgur-Bertok D.; Starčič Erjavec M. The discriminative power in determining genetic diversity of Escherichia coli isolates: Comparing ERIC-PCR with AFLP. Afr. J. Microbiol. Res. 2013, 7(20), 2416-2419, we showed that the discriminative power of ERIC-PCR in determining genetic diversity is as good as of AFLP. This paper is now cited in the revised manuscript.
Reviewer's specific comments
#Materials and Methods: number of farms, information about farms, pathologies, recovered organs, used media, number of samples, remove Table 1 into supplementary materials
The data that we can disclose is now inserted in the Bacterial strains section and Supplementary Table 1. The table 1 is removed into supplementary materials.
#Results – repetition of data in the text and in the tables, Figure 1 into supplementary materials (+ include data on the Fig1) and #Discussion – too long, literature review
The sections Results and Discussion are now completely revised and joined together into one section. Further some parts of the Discussion were moved into Introduction. Figure 1 is moved into supplementary material and the wanted data is added.
Round 2
Reviewer 2 Report
It is now clearer that this investigation is about “Escherichia coli isolated from cases of avian colibacillosis in three Russian poultry farms (Perm Krai): sensitivity to antibiotics and bacteriocins”. I understand the reluctance (and maybe the impossibility) of performing additional work. However, I am still convinced that this manuscript is not enough for accepting because both major shortages of this study remain: (i) the limited number of isolates tested and (ii) the basic investigation regarding antibiotic resistant genes.
Author Response
Dear Sir,
Thank you very much for all your work on our manuscript and for all your suggestions. We do agree that the final number of strains included in the study is rather small. However, the sample is large enough, and also based only on non-clonal E. coli isolates, to give the feeling about the situation we would like to report to the broader microbiological community. We are all aware that the number of antibiotic resistant strains is raising and that one of the hopes for alternative (protection) treatments are probiotic strains, also in Russia. But our study shows, that the Russian E. coli strains found on poultry farms have a high insensitivity to antibiotics (also based on ESBL genes), as expected, but an even higher insensitivity to bacteriocins, so that special care has to be taken when probiotics are used. Further, in the final remarks of the manuscript, it is now written that larger studies should be performed and we do hope that other microbiological groups will be inspired in testing the insensitivities to antibiotics and bacteriocins on a larger collection of APEC strains.
Sincerely,
Marjanca Starčič Erjavec, PhD